# Effect of Different N:P Ratios on the Growth, Toxicity, and Toxin Profile of *Gymnodinium catenatum* (Dinophyceae) Strains from the Gulf of California

**DOI:** 10.3390/toxins14070501

**Published:** 2022-07-18

**Authors:** Francisco E. Hernández-Sandoval, José J. Bustillos-Guzmán, Christine J. Band-Schmidt, Erick J. Núñez-Vázquez, David J. López-Cortés, Leyberth J. Fernández-Herrera, Carlos A. Poot-Delgado, Manuel Moreno-Legorreta

**Affiliations:** 1Centro de Investigaciones Biológicas del Noroeste (CIBNOR), Av. Instituto Politécnico Nacional 195 Playa Palo de Santa Rita, La Paz C.P. 23096, Mexico; dlopez04@cibnor.mx (D.J.L.-C.); legoreta04@cibnor.mx (M.M.-L.); 2Instituto Politécnico Nacional, Centro Interdisciplinario de Ciencias Marinas (IPN-CICIMAR), Av. IPN s/n, Playa Palo de Santa Rita, La Paz C.P. 23096, Mexico; cbands@ipn.mx (C.J.B.-S.); lfernandezh1200@alumno.ipn.mx (L.J.F.-H.); 3Instituto Tecnológico Superior de Champotón, Carretera Champotón-Isla Aguada Km 2, El Arenal, Champotón, Campeche C.P. 24400, Mexico; cpoot35@gmail.com

**Keywords:** *Gymnodinium catenatum*, paralytic toxins, semi-continuous culture, toxin profile, N:P ratio, Gulf of California

## Abstract

The harmful microalgae *Gymnodinium catenatum* is a unique naked dinoflagellate that produces paralytic shellfish poisoning toxins (PSTs). This species is common along the coasts of the Mexican Pacific and is responsible for paralytic shellfish poisoning, which has resulted in notable financial losses in both fisheries and aquaculture. In the Gulf of California, *G. catenatum* has been related to mass mortality events in fish, shrimp, seabirds, and marine mammals. In this study, the growth, toxin profiles, and toxin content of four *G. catenatum* strains isolated from Bahía de La Paz (BAPAZ) and Bahía de Mazatlán (BAMAZ) were evaluated with different N:P ratios, keeping the phosphorus concentration constant. All strains were cultivated in semi-continuous cultures (200 mL, 21.0 °C, 120 µmol photon m^−2^s^−1^, and a 12:12 h light-dark cycle) with f/2 + Se medium using N:P ratios of: 4:1, 8:1, 16:1, 32:1, and 64:1. Paralytic toxins were analyzed by HPLC with fluorescence detection. Maximum cellular abundance and growth were obtained at an N:P ratio of 64:1 (3188 cells mL^−1^ and 0.34 div day^−1^) with the BAMAZ and BAPAZ strains. A total of ten saxitoxin analogs dominated by N-sulfocarbamoyl (60–90 mol%), decarbamoyl (10–20 mol%), and carbamoyl (5–10 mol%) toxins were detected. The different N:P ratios did not cause significant changes in the PST content or toxin profiles of the strains from both bays, although they did affect cell abundance.

## 1. Introduction

The athecate dinoflagellate *Gymnodinium catenatum* [1] is the only naked gymnodinioid known to produce paralytic shellfish poisoning toxins (PSTs). Information of the global distribution of *G. catenatum* has increased over recent decades, and the species is currently known to be distributed worldwide across various coastal ecosystems [2,3,4,5,6,7,8,9,10,11,12]. In Mexico, *G. catenatum* has been reported along the Pacific coast, in the Gulf of California [9,13,14,15,16,17,18,19,20], and along the Campeche coast of the Gulf of Mexico [21]. On multiple occasions, *G. catenatum* has been related to shellfish contaminated with PSTs in the Mexican Pacific and the Gulf of California [9,17,18,22,23,24,25,26,27,28,29]. In some of these cases, the consumption of contaminated bivalve mollusks resulted in human fatalities. Not surprisingly, *G. catenatum* has caused notable financial losses in both fisheries and shrimp aquaculture. In addition, many epizootic diseases in fish, seabirds, sea turtles, and marine mammals have been associated with this gymnodinioid [9,13,22,23,28,29,30,31,32,33,34]. Interestingly, studies with *G. catenatum* strains isolated from Bahía de La Paz and Bahía Concepción in the Gulf of California have shown that the toxin profiles of the strains from both locations were similar [14,35]. 

The widespread proliferation of *G. catenatum* in various marine ecosystems is due to its tolerance to a wide range of salinity and temperature [16,28], in addition to its ability to form resistant cysts [6,11,36] and withstand different nutrient regimes [37]. Moreover, the growth and toxicity of dinoflagellates such as *G. catenatum* are mainly influenced by light intensity, temperature, salinity, nutrients, and N:P ratio [17,35,38,39,40,41,42,43,44,45,46].

Some studies have been conducted to determine the role that nutrients play in PSTs production, although these have been mainly conducted with *Alexandrium* spp. [38,47,48,49]. These studies have demonstrated that biochemical changes at the cellular level and a decrease in toxin content occur when the supply of N is deficient, both of which affect cell division and lead to a decrease in cell abundance. The role of phosphorus in PST synthesis is not well understood. However, some studies have shown that PST production increases while cell division decreases in culture media deficient in phosphorus [39,50,51], which suggests that a mechanism exists that allows for microalgae to accumulate nitrogen under phosphorus-deficient conditions.

In the Gulf of California, wastewater discharge, increases in coastal zone use, and other anthropogenic factors in addition to natural processes such as upwelling and river runoff, have increased the concentrations of nitrogen and phosphorus compounds over the last three decades [31,52,53,54]. For this reason, our aim in this study was to improve our understanding of the ecophysiology of *G. catenatum* by determining the effects of different N:P ratios on the cell density, growth rate, toxicity, and toxin profiles of *G. catenatum* strains isolated from Bahía de La Paz and Bahía de Mazatlán in the Gulf of California and grown in semi-continuous cultures. 

## 2. Results

### 2.1. Average Growth Rate

The growth rates of *G. catenatum* strains from both bays varied from 0.09 to 0.32 div day^−1^ (Figure 1A–D), and statistically significant differences were found among N:P ratios and strains. In the strains GCMQ-4, BAMAZ-2 and BAPAZ-5, a tendency for the growth rate to increase was observed as the N:P ratio increased from 4:1 (0.13–0.29 div day^−1^), 8:1 (0.15–0.22 div day^−1^), and 16:1 (0.22–0.29 div day^−1^). However, with higher N:P ratios of 32:1 and 64:1, the opposite was observed in BAMAZ-2 (Figure 1C). Maximum growth rates of 0.32 and 0.31 div day^−1^ were observed in BAPAZ-7 and BAPAZ-5 with the N:P ratios of 32:1 and 64:1, respectively (Figure 1B,D).

### 2.2. Average Cell Abundance in Semi-Continuous Cultures

The average cell abundance in the semi-continuous cultures varied between 700 and 3200 cells mL^−1^ (Figure 1E–H). Strain BAPAZ-5 maintained an average cell abundance of 734 ± 275 and 2444 ± 212 cells mL^−1^ with N:P ratios of 64:1 and 4:1, respectively (Figure 1F). The maximum cell abundances observed in BAPAZ-7 were recorded with N:P ratios of 64:1 and 4:1 (2530 ± 327 and 1900 ± 259 cells mL^−1^, respectively; Figure 1H). Strain BAMAZ-2 showed maximum cell abundance with N:P ratios of 64:1 and 8:1 (3188 ± 1105 and 2157 ± 347 cells mL^−1^, respectively; Figure 1G). The average density observed in GCMQ-4 ranged from 829 ± 120 to 1477 ± 393 cells mL^−1^ (Figure 1E). The ANOVA results revealed no significant differences within N:P ratios for GCMQ-4 (*p* < 0.05; Figure 1E). BAMAZ-2 showed average cell abundances that were significantly different between the N:P ratios of 64:1 and those of 4:1, 8:1, 16:1, and 32:1 (*p* < 0.05; Figure 1G). The average cell abundance of BAPAZ-5 tended to decrease as the N:P ratio increased; statistical analyses showed significant differences of the N:P 64:1 with the N:P ratios of 4:1, 8:1, 16:1, and 32:1 (Figure 1F). BAPAZ-7 showed an inverse tendency to that of BAPAZ-5, as its average cell abundance increased as the N:P ratio increased, and significant differences between the N:P ratio of 64:1 and those of N:P ratios of 4:1, 8:1, 16:1, and 32:1 were found (*p* < 0.05; Figure 1H).

### 2.3. Toxin Content

In general, the toxin content was significantly different between strains cultured at N:P ratios of 4:1, 32:1, and 64:1, with significantly greater toxin content registered in BAPAZ-7 at an N:P ratio of 64:1 with respect to those of the other strains and ratios (149.81 ± 25.54 pg STXeq cell^−1^; Figure 2E). BAMAZ-2 showed maximum toxin content of 58.45 ± 8.16 and 64.67 ± 10.14 pg STXeq cell^−1^ with the N:P ratios of 32:1 and 64:1, respectively (Figure 2D,E). An increase in toxin content was evident in BAMAZ-2 and BAPAZ-7 with the N:P ratios of 32:1 and 64:1 (Figure 2D,E), whereas BAPAZ-5 showed the inverse of this tendency (Figure 2A–E). The toxin content of GCMQ-4 varied from 4.89 to 10.54 pg STXeq cell^−1^, and statistical analyses revealed significant differences within N:P ratios (Figure 2A,B).

The toxin content (pg cell^−1^) varied between strains according to N:P ratio, and distinct differences were present in the toxin content of the *G. catenatum* strains in response to the different N:P ratios. More than 90% of the toxin content was composed of decarbamoyl and sulfocarbamoyl toxins, whereas less than 10% was attributed to carbamoyl toxins (Figure 3). GCMQ-4 presented the lowest toxin content when compared to the other strains (from 8.17 ± 0.92 to 12.34 ± 2.36 pg cell^−1^; Figure 3). In general, the toxin content of this strain did not show any clear trend during the experimental period (Figure 3). The maximum toxin content of BAMAZ-2 (28.65 ± 4.36 pg cell^−1^) was registered with the N:P ratio of 4:1, and the ANOVA results did not indicate any significant differences within the different N:P ratios of 16:1, 32:1, and 64:1 (28.21 ± 5.12, 24.98 ± 6.21, and 18.76 ± 4.78 pg cell^−1^, respectively). BAPAZ-5 and BAPAZ-7 showed maximum toxin content of 51.54 ± 8.45 and 38.26 ± 6.12 pg cell^−1^ with the N:P ratios of 16:1 and 64:1, respectively (Figure 3). The toxin content of BAPAZ-5 showed a clear trend during the experimental period, as its toxin content increased when the N:P ratio increased from 4:1 (12.32 ± 2.36 pg cell^−1^) to 16:1 (51.54 ± 8.45 pg cell^−1^). In comparison, BAPAZ-7 showed higher toxin content with N:P ratios ranging from 16:1 (25.48 ± 5.98) to 64:1 (38 ± 7.34 pg cell^−1^; Figure 3).

### 2.4. Toxin Profile

The toxin profiles of *G. catenatum* show the presence of ten STX analogs from the sulfocarbamoyl (B1, B2, C1, and C2), decarbamoyl (dcSTX, dcNEO, dcGTX2, and dcGTX3) and carbamoyl (GTX2 and GTX3) toxin groups (Figure 4). The molar percentages recorded for GCMQ-4 shown in Figure 5 indicate a high molar percentage of sulfocarbamoyl toxins (45–88 mol%), followed by those of decarbamoyl toxins (0.87–36 mol%) and carbamoyl toxins (5.02–18 mol%). The results of the statistical analyses (mol%) indicated that significant differences were found among all N:P ratios (*p* ˂ 0.005) in the three toxin groups. Strain BAMAZ-2 tended to increase its molar percentage of sulfocarbamoyl toxins from 60 to 90 mol% as the N:P ratio increased from 4:1 to 64:1. However, the tendency of decarbamoyl and carbamoyl toxins to increase was inverse to that of the sulfocarbamoyl toxins (Figure 5). In BAPAZ-5, the molar percentage of sulfocarbamoyl toxins decreased (96 to 78 mol%) while that of the decarbamoyl toxins increased (3 to 22 mol%) as the N:P ratio increased (Figure 5). In BAPAZ-7, a decrease in the molar percentage of sulfocarbamoyl toxins (90 to 71 mol%) was observed while that of decarbamoyl toxins increased (2 to 30 mol%) as the N:P ratio increased from 16:1 to 64:1. Carbamoyl toxins did not show significant changes in molar percentage in any strain (Figure 5).

## 3. Discussion

The present study compared the growth rates, toxin content, and PST profiles of four strains of *G. catenatum* obtained from Bahía de La Paz and Bahía de Mazatlán cultivated with different N:P ratios in semi-continuous cultures. When comparing growth rates and average maximum abundance among *G. catenatum* strains from the Gulf of California grown under different culture conditions to those obtained in this study, we observed similar results (Table 1). In particular, we can observe that the growth rates recorded in this study (0.21–0.34 div day^−1^) are similar to those reported by other authors for *G. catenatum* strains grown under similar culture conditions (0.19–0.24 div day^−1^) [56] using the same culture media [28,57]. However, previous studies have reported higher growth rates (0.57–0.8207 div day^−1^) using different culture media, such as GSe or f/2 with Se and soil extract, than those in this study (see Table 1) [45,58,59,60,61]. The differences in the growth rates among strains from the same region could be due to the different culture media used in each experiment and the methods employed to isolate each strain. Han et al. [10] mention that the growth rates of *G. catenatum* from Korean waters can vary between strains and may depend on the geographical origin of each strain. Kim et al. [62] mentioned that nitrogen is one of the most critical nutrients that affects cell growth and the biochemical composition of microalgae.

The role of nutrients (primarily nitrogen and phosphorus) in the growth of *G. catenatum* strains from the Gulf of California in semi-continuous regimes remains poorly studied. However, the function of nitrogen in the physiology of dinoflagellates has been reviewed and described by some authors such as Dagenais-Bellefuille and Morse [62], and its particular importance has been highlighted both in primary metabolism, as in the synthesis of amino acids, nucleic acids, and chlorophyll (e.g., photosynthesis), and secondary metabolism in the production of various phycotoxins (e.g., PSTs biosynthesis). Thus, changes in the concentrations of various nitrogenous compounds can significantly affect these physiological processes. These authors describe that a high concentration of nitrogen can be correlated with an increase in cell division that results in the formation of harmful algal blooms (HABs), whereas decreases in nitrogen can cause cell cycle arrest, and induce physiological, behavioral, and transcriptomic changes. Phosphorus is also an essential macronutrient needed to form compounds that are later used in energy transport, such as ATP, ADP, NADPH, and NADP molecules [62]. Thus, these nutrients are components of cell membranes in the form of compounds such as phospholipids, which phytoplankton incorporate directly.

In this study, the high nitrogen concentration of 232 µM (N:P of 64:1) was related to high cell abundance in BAPAZ-7 (3983 ± 375 cell mL^−1^) and BAMAZ-2 (4000 ± 276 cell mL^−1^; Appendix A). Han et al. [10] found that *G. catenatum* showed the lowest cell density (485 cells mL^−1^) with the high N:P ratio of 48:1, indicating that N:P ratios above 24:2 did not increase the growth rate or cell abundance of strains isolated from Korea. 

These results indicate that it is not possible to identify a pattern with regard to the effects of different N:P ratios on the variables measured in this study. However, statistically significant differences were identified between strains. For example, BAPAZ-5 showed better growth at the lowest N:P ratios when compared to the growth of other strains in this study (Figure 1B), although the opposite was observed with the growth of BAMAZ-2. Collectively, the variability among strains and the absence of a clear pattern, either among strains of the same region or among different geographic regions, has been associated with the genetic diversity of phytoplankton species. According to Wood and Leatham [65], this diversity is responsible for the contrasting results observed when strains of the same species are evaluated. In this study, we used only two strains from each bay, BAPAZ-5 and BAPAZ-7 from Bahía de La Paz and BAMAZ-2 and GCMQ-4 from Bahía de Mazatlán. Both BAPAZ-5 and GCMQ-4 were isolated in 2000, while the other strains were isolated in June 2006 and May 2007. The method used to isolate *G. catenatum* strains may be one factor responsible for the aforementioned differences. In addition, the time of isolation may also be in part responsible for the differences found among strains in this study [65]. BAPAZ-5, BAPAZ-7, and BAMAZ-2 were isolated from vegetative cells, while GCMQ-4 was established by cyst germination. 

We confirmed the presence of ten PSTs in *G. catenatum* and detected N-sulfocarbamoyl (C1, C2, B1, and B2), decarbamoyl (dcSTX, dcNEO, dcGTX2, and dcGTX3), and carbamoyl (GTX2 and GTX3) toxins (Figure 4). Nevertheless, the presence of STX derivatives, such as the benzoates described for this species [66,67], cannot be discarded because the method we used does not detect them. In previous studies with *G. catenatum* strains from the Mexican Pacific, the presence of five hydroxy-benzoyl analogues (GC 1/2, GC3, and GC4/5) and two sulfated benzoyl analogues (GC 1b/2b) was confirmed [35]. This toxin profile is similar to those of previous reports for strains from the same area [14,57,58]. Band-Schmidt et al. [57] suggested that the toxin profiles (dcSTX, dcGTX2/3, and C1/2 as well as low proportions of GTX2/3 and B2) could be used as biomarkers for strains from Bahía Concepción, in the Gulf of California. However, differences among toxin profiles can also be observed among different culture media and culture lengths [17,58]. The dcNEO toxin was always present in the four strains of *G. catenatum* cultured in f/2 media; however, dcNEO has not yet been reported for strains from the Gulf of California [14,17,58]. Bustillos-Guzmán et al. [35] mentioned that the erroneous identification of peaks in fluorescence (false positives) was due to the detection method used and that the recurrent finding of NEO in strains from the Gulf of California corresponds to dcNEO, as confirmed by LC-MS/MS.

Studies related to the toxin content of *G. catenatum* in semi-continuous cultures are scarce. Béchemin et al. [68] cultured the dinoflagellate *Alexandrium minutum* at five N:P ratios and observed an increase in the toxicity per cell (1.24–8.01 fmol cell^−1^) in cultures with N:P ratios ranging from 1.16:1 to 80:1, observing a significant increase in cellular toxicity under phosphorus-limited conditions (i.e., N:P 80). We observed a similar trend in BAMAZ-2 (12 to 60 pg STXeq cell^−1^) and BAPAZ-7 (32 to 158 pg STXeq cell^−1^; Figure 2). The inverse of this pattern was observed with GCMQ-4, with low toxicity (12 to 3 pg STXeq cell^−1^) under nitrogen-deficient conditions. GCMQ-4, obtained from a cyst germination, showed the lowest toxicity of the four strains in this study. The other three strains, which were isolated from vegetative cells, increased their toxin content up to 6-fold compared to that of GCMQ-4. Differences in the toxicity and toxin profiles between strains isolated from cysts and vegetative cells have been reported. Oshima et al. [69] observed that the isolation method could be related to the toxin content at different N:P ratios. Negri et al. [66] observed that cultures generated from cyst germination usually exhibited 10-fold lower toxin content that cultures isolated from vegetative cells. These differences found in the content of toxins may be due to the influence of bacteria at the time of germination of the cyst. Green et al. [70] suggest that bacteria actively participate in the production of PSTs. They hypothesized that bacteria are required to activate the biosynthesis of PSTs of *G. catenatum* and that the process of cleaning (sonication and washing) of cysts in sterile seawater as part of the process of isolation of cysts from sediments may cause the observed reduction in PSTs. We observed a clear pattern in three of the four strains in this study, with toxin content decreasing or increasing in treatments in which nitrogen or phosphorus was limiting, respectively (Figure 2A–E). These results may indicate that nitrogen plays a functional role in the biosynthesis of PSTs. 

Amino acids rich in N, such as arginine, and lysine are precursors to the synthesis of PSTs, and thus an increase in nitrogen could increase the synthesis of these precursors and consequently the toxin content in cells [71,72]. John and Flynn [73] found a positive relationship between toxin content and the intracellular concentration of arginine in *Alexandrium fundyense*. Anderson et al. [39] analyzed different metabolites produced by *A. fundyense* under phosphorus-limiting culture conditions and detected an increase in the concentration of arginine, the precursor of saxitoxin. This was also shown by John and Flynn [73] with the same species. When phosphorus is added to deficient media, cell division increases and the concentration of toxins per cell decreases [74,75].

Limited variation was observed in the toxin content per cell among the four strains in semi-continuous cultures and different N:P ratios in this study, with values ranging from 4.8 to 38 pg cell^−1^ (Figure 3). Bustillos-Guzmán et al. [56] also did not observe significant changes in toxin content when varying the nutrient concentration in batch cultures of *G. catenatum* and attributed these results to the short acclimatization period of the strains. However, we acclimatized the strains in this study four cell cycles to each N:P ratio before initiating the experiment; therefore, the acclimatization of the strains to the different N:P ratios did not have a notable effect on the cell content, as shown by our results (Figure 3). Wide variation in toxin content per cell has been reported for *G. catenatum* strains isolated from the same geographic area, and this may be considered a common characteristic [58].

On a molar basis, as is characteristic for strains from the Gulf of California, the sulfocarbamoyl toxin C1/2 and decarbamoyl toxin dcGTX2/3 contributed the highest molar percentages to total toxicity (Figure 5) [17,35]. GCMQ-4 showed an increase in the molar percentage of carbamoyl toxins from 4 to 32 mol% as the N:P ratio increased (Figure 5A–D) along with a decrease in the molar percentage of sulfocarbamoyl toxins (48 mol%); however, this strain was the least toxic. Similar results have been previously reported for other strains isolated from Bahía de Mazatlán [17]. 

The effects of different nitrogen and phosphorous concentrations on growth, toxin content, and toxin profiles have been evaluated in other species and not only in PST-producing dinoflagellate species, such as *Alexandrium* spp, *Pyrodinium bahamense* and *Gymnodinium catenatum.* The cell toxin content of these species has been found to change according to the N:P ratio and culture media (Table 2) [10,47,56,57,58,68,72,73,76,77,78,79,80]. In addition, these factors have also been found to influence the growth of other harmful microalgae species, such as the dinoflagellates *Karenia mikimotoi*, *Prorocentrum donghaiense* [44], *Amphidinium carterae* [46], *Prorocentrum minimum* [81], *Prorocentrum lima*, and *Dinophysis* spp., the production of toxins (e.g., diarrhetic toxins) and other physiological processes [82,83,84]. Similar observations have been made with brevetoxin production in *Karenia brevis* [85,86] and in the production of some ciguatoxin analogs by *Gambierdiscus* spp. [87,88]. In addition, the N:P ratio and growth medium have also been found to affect the growth of *Ostreopsis* c. f. *ovata* and its production of ostreocins [89,90], *Margalefidinium polykrikoides* [91], and the raphidophyte *Fibrocapsa* [92]. Domoic acid production in diatoms of the genus *Pseudo-nitzschia* has been found to be influenced by phosphorus and nitrogen deficiency [93]. Cyanobacteria growth [94] and the production of cyanotoxins, such as cylindrospermopsins and microcystins, by *Raphidiopsis raciborskii*, *Microcystis* sp., and *Planktothrix* sp., have also been found to be affected by nitrogen and phosphorous concentration [73,95,96,97]. 

There is a consensus among some research groups regarding the effect that eutrophication can have on the global increase of harmful algal blooms [43], namely that higher inputs of key nutrients, such as nitrogen and phosphorus, can stimulate the growth of harmful microalgae species. Some species of dinoflagellates and cyanobacteria are harmful because they produce N-rich toxins (e.g., PSTs, cylindrospermopsin, microcystin, and Nodularin, with C:N ratios of 1.5, 3.0, 4.3, and 5.1, respectively) that harm humans, animals, and the overall health of ecosystems [72,98]. Identifying quantitative relationships between nutrient inputs and the proliferation of specific microalgae is challenging and complex due to the diversity of the sources, forms, and flows of both exported and cycled nutrients; the diversity of algal nutrient acquisition mechanisms; and the interactions among noxious species and other organisms within food webs [99]. 

Some studies have used meta-analyses to evaluate how changes in nutrient availability can influence the synthesis of N-rich phycotoxins. For example, Branderburg et al. [72] described that N-rich phycotoxin content generally increased or decreased under conditions of phosphorus limitation or nitrogen limitation, respectively, although variation among responses both within and between toxin-producing genera and toxin analogs was present. Both the production and composition of phycotoxins have been shown to follow stoichiometrically predictable patterns [98,100]. However, Davidson et al. [100] indicated that if the nutrient concentrations were not limiting for growth, then the nutrient ratio would not influence the resulting floristic composition. At non-limiting concentrations, evidence that changing N:P ratios stimulate HABs is limited and based primarily on hypothetical relationships derived from the data of only a few sites. In all cases, an unequivocal causal link between increases in the frequency, magnitude, or duration of HABs and changes in either nitrogen or phosphorus limitation have been difficult to establish. Bellefeuille and Morse [62] mentioned that nitrogen is generally required for the synthesis of amino acids, nucleic acids, chlorophylls, and toxins, and these changes in nitrogen concentration strongly affect cell density, primary production, and secondary metabolism. For example, high nitrogen concentrations are correlated with high cell division, resulting in the formation of HABs [62].

In vitro studies have shown that nutrient ratios can influence phycotoxin production, but sex- and species-based differences and controlled environmental conditions make it difficult to extrapolate results to explain what may happen in natural settings [100]. Our understanding of the roles that particulate and dissolved organic nutrients (especially organic nitrogen) play in the growth of harmful microalgae remains limited, although these roles may be quite relevant. A better understanding of the role of mixotrophy in HAB formation and of the competition for environmentally realistic concentrations of organic nutrients between HAB-forming and non-HAB-producing species is needed [100,101].

Our results with four isolated strains of *G. catenatum* from Bahía de La Paz and Bahía de Mazatlán under the conditions of this study coincide in this sense, as the different N:P ratios did not notably affect the variation in growth rates, toxin profiles, or toxicity. However, growth rates and cell abundance were influenced to the greatest extent by the specific concentrations of the nitrogen source (NaNO_3_).

## 4. Conclusions

The different N:P ratios in semi-continuous cultures did not significantly affect the toxin profiles or the toxin content in the BAPAZ and BAMAZ strains of *G. catenatum*. Nevertheless, the growth rates and cellular abundance were influenced by the N:P ratios using NaNO_3_ as the nitrogen source. These finding highlights the wide range of N:P ratios in which *G. catenatum* could grow under natural conditions. However, analyses of other regions of the Gulf of California are needed, using other sources of nitrogen and determining metabolites such as arginine as a precursor in the biosynthesis of PSTs to understand how the N:P ratio and nitrogen concentration affect both growth rate and toxin content.

## 5. Materials and Methods

### 5.1. Strains

Two strains from Bahía de La Paz (BAPAZ-7 and BAPAZ-5) and two strains from Bahía de Mazatlán (BAMAZ-2 and GCMQ-4) were used in this study (Figure 6). Table 3 shows the isolation information for each strain. Strain GCMQ-4 (Bahía de Mazatlán, Sinaloa) was acquired from the marine dinoflagellate collection (CODIMAR) of CIBNOR in La Paz, Mexico. The isolation information of the strains in this collection can be found at https://www.cibnor.gob.mx/investigacion/colecciones-biologicas/codimar accessed on 13 July 2022.

Vegetative cells were collected by vertical and superficial trawls with a plankton net (20 μm mesh). The concentrated phytoplankton collected in each trawl was passed through a 60 μm mesh net to remove larger organisms. The filtrate was inoculated in 250 mL containers with f/2 culture medium [106] that had been modified by adding selenium (10^−8^ M H_2_SeO_3_) and reducing the copper concentration (10^−8^ M CuSO_4_) [107]. In the laboratory, vegetative cells of *G. catenatum* (Appendix A) were isolated by single cell isolation with a capillary tube under an Axiovert 100 inverted microscope (Carl Zeiss, Oberkochen, Germany) and individually transferred to a 24-cell plate containing the modified f/2 medium. Cells were incubated at a temperature of 21.0 ± 1.0 °C, light intensity of 120 μmol photon m^−2^ s^−1^, and a 12:12 h light–dark cycle.

### 5.2. Experimental Culture Condition of Gymnodinium Catenatum

The growth of the four *G. catenatum* strains was followed in 500 mL flasks with 250 mL of culture medium. The strains were first grown with an N:P ratio of 16:1 to determine the sampling times and growth phases using NaNO_3_ as the nitrogen source (Appendix A). Every third day, a 1 mL sample was obtained in triplicate for cell counts, and the samples were fixed with acid Lugol [108]. Cell counts were performed in a Sedgewick-Rafter chamber (1-mL capacity) with a DMLS optical microscope (LEICA, Wetzlar, Germany).

### 5.3. Experimental Design

For experimental purposes, batch and semi-continuous cultures were used. Batch cultures were performed with the four *G. catenatum* strains in triplicate in 500 mL Erlenmeyer flasks. Growth rates and toxin concentrations were evaluated using NaNO_3_ as the nitrogen source. The culture media were prepared by varying the amount of nitrogen, keeping the phosphorus concentration constant (3.63 µM) in the medium, and five N:P ratios of 4:1, 8:1, 16:1, 32:1, and 64:1 were used with concentrations of 14.5:3.63, 29:3.63, 58:3.63, 116:3.63, and 232:3.63 μM, respectively. These N:P ratios were selected according to those found in both bays, starting with the N:P ratio 16:1 and increasing and decreasing the nitrogen concentration from this ratio by an order of 2.

Flasks were inoculated with 10% of the volume of the stock culture with cell abundances that ranged between 300 and 800 cells mL^−1^ in 200 mL of culture medium. Cultures were progressively conditioned to each N:P ratio from the initial condition (16:1). For each N:P ratio, the cells were adapted for at least three generations before starting the semi-continuous cultures. Once the cultures were conditioned in each of the N:P ratios, growth was determined as described in Section 5.2. 

Semi-continuous cultures were initiated at the beginning of the stationary phase. The cultivation system was modified from a batch to a semi-continuous culture system, once again conditioning all strains for two generations at each N:P ratio. In the semi-continuous system, 2 mL samples were collected for cell counts and 30 mL samples were collected for PST analysis every second day for 10 days. Samples intended for cell counts were fixed with acid Lugol, while the samples intended for PST analysis were concentrated by filtration. Approximately 20% of the *G. catenatum* culture media was replaced with ~40 mL of fresh media after sampling. The culture conditions were the same as those described above.

### 5.4. Cell Growth

To determine the cell abundance and the growth rate, 10% of the volume of the stock culture was inoculated with ~300–800 cells mL^−1^ in 200 mL of culture medium. Cell abundance was estimated by collecting 2 mL of the culture media and fixing it with acid Lugol every third day and then counting cells until the decay phase was identified [108]. The counts were performed in a 1 mL Sedgwick-Rafter chamber with a LEICA DMLS optical microscope. The growth rates were calculated using the following Equation (1) [109]:(1)$$\mu =\frac{\left(lnX_{t}-lnX_{0}\right)}{t}$$
were *X*_0_ is the initial cell density, *X_t_* is the cell at time *t*, and *t* is the time in days.

### 5.5. HPLC-FLD Analysis of PSTs

Samples were extracted with 1 mL of acetic acid 0.03 N. Each sample was sonicated three times for 1 min. The crude extract was centrifuged at 14,000 rpm (18,502.4× *g*) (HERMLE Z 216 microcentrifuge, Labortechnik GmbH, Wasserburg, Germany) for 15 min at 10 °C. The supernatant was passed through a syringe (25 mm diameter) with glass fiber filters (0.45 μm pore size; PVDF Millex membrane). For hydrolysis, 150 μL of each extract was mixed with 37 μL of HCl 1N and heated with a headblock (VWR digital heatblock, Troemner, LLC, Thorofare, NJ, USA) for 15 min at 90 °C to convert the N-sulfocarbamoyl toxins to their corresponding carbamated toxins. After cooling at room temperature, the samples were neutralized with 75 μL of CH_3_COONa 1N. The extracts were stored at −20 °C until analysis (before 24 h). The hydrolyzed and non-hydrolyzed extracts were injected into an HPLC liquid chromatograph in independent runs to identify and quantify previously reported PSTs [110,111]. N-sulfocarbamated toxins (including GTX5/B1 and GTX6/B2) were quantified by calculating the maximum peak increase, which was used to relate them to the carbamated toxins formed during the treatment with HCl (B1 to STX, B2 to neoSTX, C1 to GTX2, C2 to GTX3, C3 to GTX1, and C4 to GTX4). An HP Agilent 1100 chromatography system (Santa Clara, USA) was used, which consisted of an autosampler, degasser, quaternary pump, two binary pumps used for post-column reactions, a fluorescence detector, a C-18 column, and a post-column reactor. The PSTs were detected using an excitation wavelength of 333 nm and an emission wavelength of 390 nm. The identification of PSTs was accomplished by comparing the retention times among samples and by coelution with commercial standards of saxitoxin (STX), neosaxitoxin (neoSTX), goniautoxin-1,4 (GTX 1,4), decarbamoylsaxytoxin (dcSTX), decarbamoylgoniautoxin-2,3 (dc GTX 2,3), and N-sulfocarbamoyl-11-hydrosulfate (C1,2; National Research Council Canada, Halifax, NS Canada).

### 5.6. Statistical Analysis

Descriptive statistics of the variables (i.e., toxin content, toxin profiles, and growth rates) were generated. A Kolmogorov–Smirnov test was used to evaluate normality, and Levene’s test was used to examine homogeneity of variance. A one-way analysis of variance (ANOVA) was used to evaluate the effect of the N:P ratio on the toxicity and growth of the naked dinoflagellate *G. catenatum*. All statistical tests were performed in Statistica v. 8.0 (Statsoft, Tulsa, AK USA). The results of the statistical tests were considered significant with a confidence level equal to or greater than 0.95 (*p* ˂ 0.05).

## Figures and Tables

**Figure 1 toxins-14-00501-f001:**
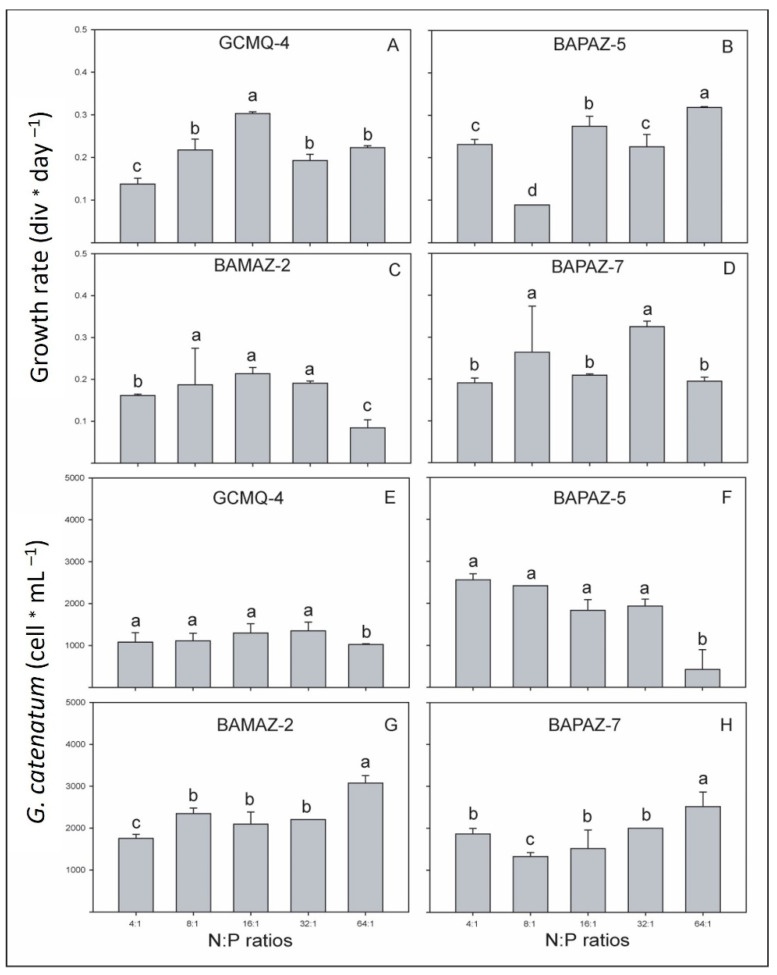
Average growth rates (**A**–**D**) and cell abundance (**E**–**H**) of *Gymnodinium catenatum* strains (i.e., GCMQ-4, BAMAZ-2, BAPAZ-5, and BAPAZ-7) grown with different N:P ratios in semi-continuous cultures. Data were analyzed using a one-way ANOVA for each strain between the five N:P ratios followed by Tukey’s post-hoc test. Different letters above the bars indicate statistically significant differences at *p* ˂ 0.05. Bars represent the standard error.

**Figure 2 toxins-14-00501-f002:**
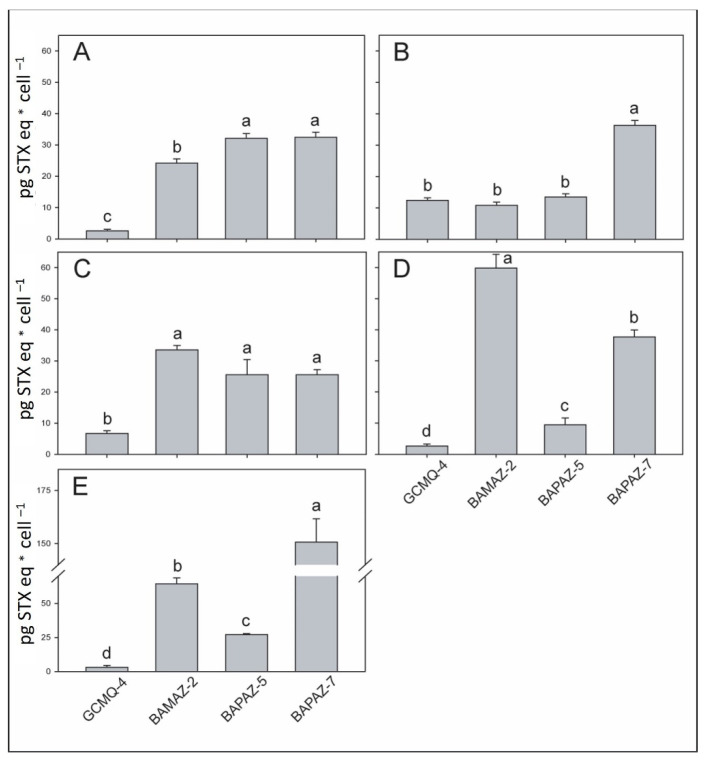
Toxin content (pg STXeq cell^−1^) in *Gymnodinium catenatum* strains (i.e., GCMQ-4, BAMAZ-2, BAPAZ-5, and BAPAZ-7) in semi-continuous cultures with N:P ratios of 4:1 (**A**), 8:1 (**B**), 16:1 (**C**), 32:1 (**D**), and 64:1 (**E**). Data were analyzed using a one-way ANOVA for each N:P ratio between strains followed by Tukey’s post-hoc test. Different letters above the bars indicate statistically significant differences at *p* ˂ 0.05. Bars represent the standard deviation.

**Figure 3 toxins-14-00501-f003:**
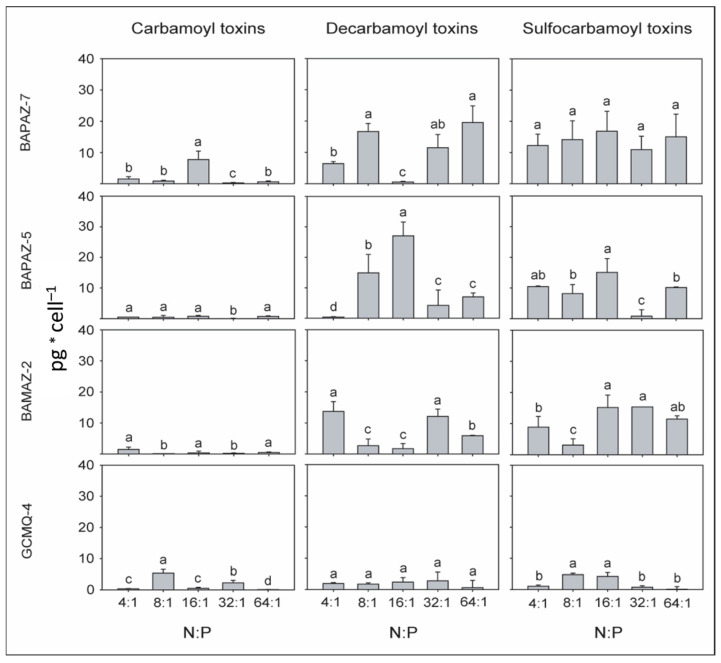
Toxin content (pg cell^−1^) in *Gymnodinium catenatum* strains (i.e., GCMQ-4, BAMAZ-2, BAPAZ-5, and BAPAZ-7) by analog (i.e., carbamoyl, decarbamoyl, and sulfocarbamoyl) for the N:P ratios of 4:1, 8:1, 16:1, 32:1, and 64:1 in semi-continuous cultures. Gray columns indicate carbamoyl (GTX 2 + GTX 3), decarbamoyl (dcSTX + dcNeo + dcGTX 2 + dcGTX 3), and sulfocarbamoyl (B1 + B2 + C1 + C2) toxins, respectively. Data were analyzed using a one-way ANOVA for each strain with five N:P ratios for each toxin group followed by Tukey’s post-hoc test. Different letters above the bars indicate statistically significant differences at *p* ˂ 0.05. Bars represent the standard deviation.

**Figure 4 toxins-14-00501-f004:**
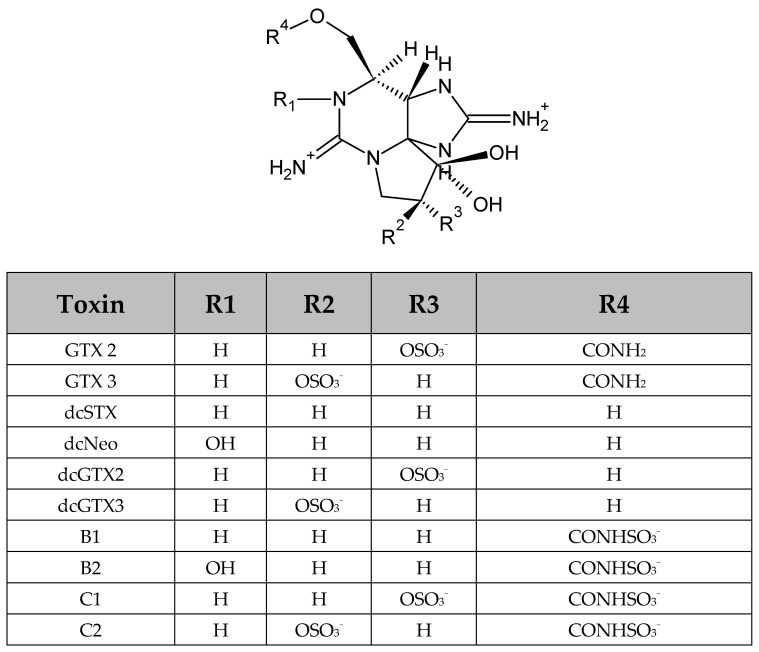
Paralytic shellfish poisoning toxins (PSTs) detected in four strains of *Gymnodinium catenatum* (i.e., BAPAZ-5, BAPAZ-7, BAMAZ-2, and GCMQ-4) collected in Bahía de La Paz and Bahía de Mazatlán. Table modified from [55].

**Figure 5 toxins-14-00501-f005:**
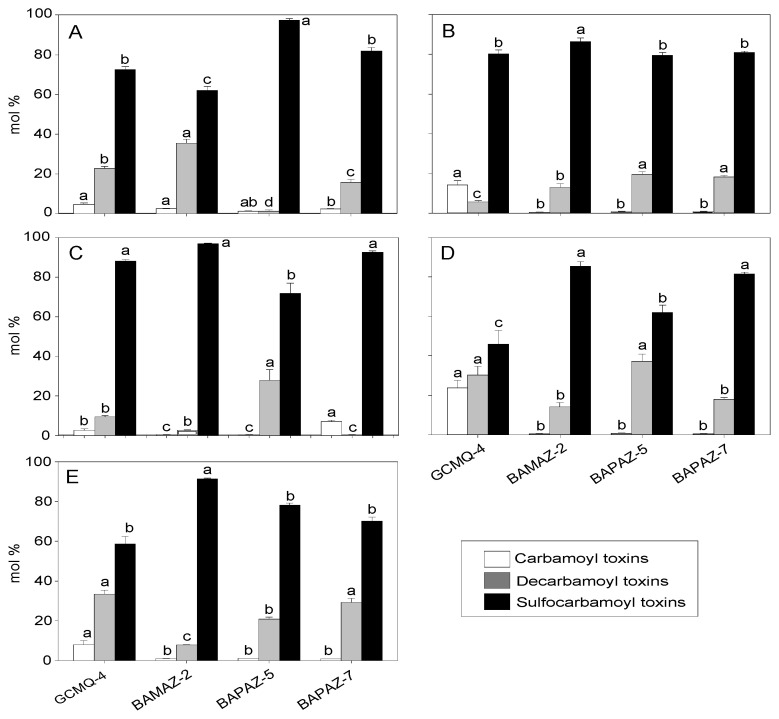
Toxins (mol%) in *Gymnodinium catenatum* strains (i.e., BAPAZ-5, BAPAZ-7, BAMAZ-2, and GCMQ-4) by analog (i.e., carbamoyl, decarbamoyl, and sulfocarbamoyl) for N:P ratios of 4:1 (**A**), 8:1 (**B**), 16:1 (**C**), 32:1 (**D**), and 64:1 (**E**) in semi-continuous cultures. White, gray, and black columns indicate carbamoyl (GTX 2 + GTX 3), decarbamoyl (dcSTX + dcNeo + dcGTX 2 + dcGTX 3), and sulfocarbamoyl (B1 + B2 + C1 + C2) toxins, respectively. Data were analyzed using a one-way ANOVA for each N:P ratio for each toxin group between strains followed by Tukey’s post-hoc test. Different letters above the bars indicate statistically significant differences at *p* ˂ 0.05. Bars represent the standard deviation.

**Figure 6 toxins-14-00501-f006:**
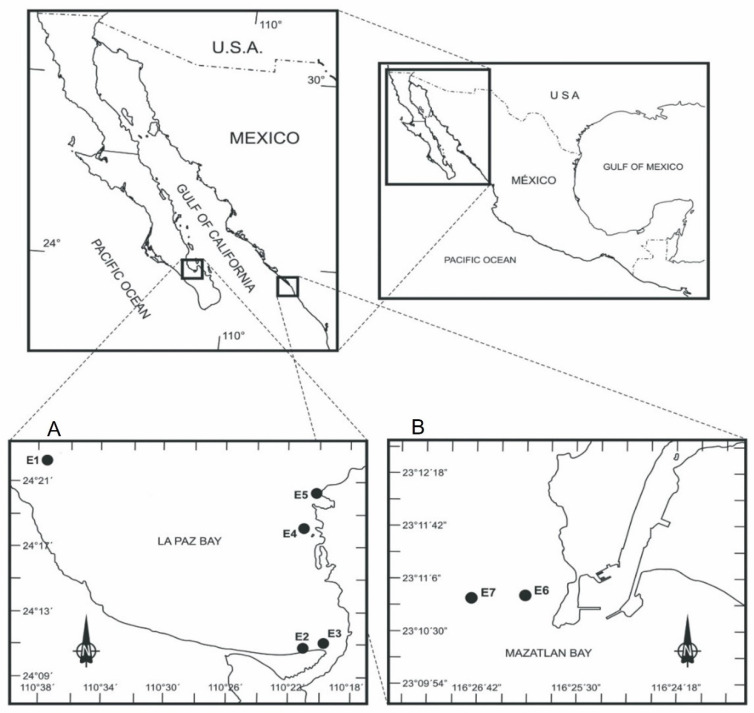
Study sites and locations of the sampling stations in (**A**) Bahía de La Paz (E1–E5) and (**B**) Bahía de Mazatlán (E6,E7).

**Table 1 toxins-14-00501-t001:** *Gymnodinium catenatum* strains isolated from Bahía de La Paz (BAPAZ), Bahía de Mazatlán (BAMAZ), and Bahía Concepción (BACO) in the Gulf of California, Mexico (Mx).

Strain Code	Origin	Growth Rate (div day^−1^)	Maximum Density (Cells mL^−1^)	Toxin Content (pg STXeq Cell^−1^)	Growth Conditions	References
GCCV-10	BACO, Mx	0.14–0.21 0.24 0.28–0.31 0.15–0.19	Nd	Nd	f/2, 15–29 °C, and salinity of 30. f/2, 20 °C, and salinity from 26 to 30. f/2, 20 °C, and salinity from 28 to 38. f/2 with Se (10^−6^ M, 10^−7^ M 10^−8^ M) and Gse, 20 °C, and salinity of 35.	Band-Schmidt et al. [28]



GCCQ-1	BACO, Mx	0.74 ± 0.07	1619 ± 252	13.16	GSe with Se concentration (10^−9^, 10^−7^ M) and salinity of ~35.	Band-Schmidt et al. [58]
GCCV-2	BACO, Mx	0.70 ± 0.07	1090 ± 270	16.63		Gárate-Lizárraga et al. [14]
GCCV-4	BACO, Mx	0.82 ± 0.09	3393 ± 836	11.66		
GCPV-1	BAPAZ, Mx	0.74 ± 0.06	1631 ± 152	12.68		
GCPV-2	BAPAZ, Mx	0.77 ± 0.05	1421 ± 290	23.29		
GCMV-1	BAMAZ, Mx	0.81 ± 0.02	2063 ± 226	23.96		
GCMV-2	BAMAZ, Mx	0.82 ± 0.03	1865 ± 516	16.60		
	BACO, Mx			60.3	f/2 + H_2_SeO_3_ (10^−8^ M)	Palomares-García et al. [63]
GCCV-7	BACO, Mx	0.19–0.24	5852	21.8	f/2 with Se (10^−8^ M), salinity of 35 with different N:P ratios (5.4, 9.2, 23.5, 44.7 y 74.3), and batch cultures.	Bustillos-Guzmán et al. [56]
BAPAZ-10	BAPAZ, Mx	0.57	4048 ± 440	370	Gse with Se (10^−7^ M) with vermicompost, salinity of 34, in batch cultures.	Fernández-Herrera et al. [61] Fernández-Herrera et al. [64] (In review)
GCMQ-4	BAMAZ, Mx	0.29 ± 0.024	2273 ± 913	3–10	f/2 with Se (10^−8^ M), salinity of 35 with different N:P ratios (4, 8, 16, 32, and 64), in semi-continuous cultures.	This study
BAMAZ-2	BAMAZ, Mx	0.21 ± 0.096	4000 ± 276	9–58	f/2 with Se (10^−8^ M), salinity of 35 with different N:P ratios (4, 8, 16, 32, and 64), in semi-continuous cultures.	This study
BAPAZ-5	BAPAZ, Mx	0.32 ± 0.035	3666 ± 798	8–32	f/2 with Se (10^−8^ M), salinity of 35 with different N:P ratios (4, 8, 16, 32, and 64), in semi-continuous cultures.	This study
BAPAZ-7	BAPAZ, Mx	0.34 ± 0.057	3983 ± 375	24–149	f/2 with Se (10^−8^ M), salinity of 35 with different N:P ratios (4, 8, 16, 32, and 64), in semi-continuous cultures.	This study

Nd. No data available.

**Table 2 toxins-14-00501-t002:** Location, toxin content, and N:P ratios of harmful algal bloom (HAB) species that produce paralytic shellfish toxins (PSTs), namely *Alexandrium* spp., *Gymnodinium catenatum*, and *Pyrodinium bahamense*.

Species/Strains	Location	Toxin Content (fmol/Cell)	Media and N:P ratio	References
*Alexandrium*: *A. affine*	Vietnam	1–2.28	Nd	Nguyen-Ngoc [102]
*A. catenella*/ACT03	Thau, France	2.90–50.3	f/2 & Provasoli	Laabir et al. [103]
*A. fundyense*/CCMP	Bigelow, Laboratories, USA	1–80 mM STX-eq	0.01: 55.55 0.18:5.42	John and Flynn [104]
*A. minutum*	Bay Morlaix, France	0.41–8.01	5.0^−3^:198.7 2:0.5	Béchemin et al. [68]
*A. peruviamum*/ApKS01	Malaysia	0.25–0.75	31.7:0.03	Lim and Ogata [41]
*A. tamarense*	Southeast, China	8–55	f/2 1.29:0.77 f/2	Wang et al. [76]
*A. tamiyavanichi*	Japan	Nd	f/2 1. 29:0.77	Oh et al. [105]
*Gymnodinium catenatum*/GCCV-6, -7, -8, -9, -10, -11, -12, -13, -14, -15, -16, -17, -18, -19, -20, -21 and -22	Mexican Pacific	26–28 pg STXeq/cell	f/2 1. 29:0.77	Band-Schmidt et al. [57]
*Gymnodinium catenatum*/GCCQ-1, GCCV-2, GCCV-4, GCPV-1, GCPV-2, GCMV-1, GCMV-2	Mexican Pacific	13–101 pg STX eq/cell	GSe 10:0.1	Band-Schmidt et al. [58]
*Gymnodinium catenatum*	Australia	Nd	7.60, 33.6, and 3.40 µmol L^−1^	Hallegraef et al. [7]
*Gymnodinium catenatum*	Korea	Nd	0.1:80.8 and 4:0.25	Han et al. [10]
*Gymnodinium catenatum*/ GCMQ-4, BAMAZ-2, BAPAZ-5, BAPAZ-7	Mexican Pacific	3–10, 9–58, 8–32, and 24–149 pg STX eq/cell	f/2 with Se (10–8 M), salinity 35 with different relations N:P (4, 8, 16, 32, and 64)	In this study
*Pyrodinium bahamense*/PbSA01	Malaysia	400	N:P-R Provasoli	Usup et al. [78]
*Pyrodinium bahamense* var. *bahamense*	Florida, USA	Nd	0.4:23.3	Phlips et al. [79]
PBC-M2-06159	Bambay Bay, Philippines	54–298	f/2 1.29:0.77	Gedaria et al. [80]

Nd. no data available.

**Table 3 toxins-14-00501-t003:** *Gymnodinium catenatum* strains isolated from Bahía de La Paz and Bahía de Mazatlán in the Gulf of California, Mexico.

Strain (Code)	Place and Date of Isolation	Water Temperature (°C)	Isolated by
BAPAZ-5	Bahía de La Paz, B.C.S., February 2007	20–21.9	C. Band-Schmidt
BAPAZ-7	Bahía de La Paz, B.C.S., February 2007	20–21.9	C. Band-Schmidt
BAMAZ-2	Bahía de Mazatlán, Sinaloa, February 2007	* Nd	C. Band-Schmidt
GCMQ-4	Bahía de Mazatlán, Sinaloa.	* Nd	L. Morquecho-Escamilla

* Nd: no data available.

## Data Availability

Not applicable.

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
