# Peer review of "Effect of Different N:P Ratios on the Growth, Toxicity, and Toxin Profile of Gymnodinium catenatum (Dinophyceae) Strains from the Gulf of California"

_toxins, 2022, doi:10.3390/toxins14070501_

Round 1
Reviewer 1 Report
Your study is interested but suffers according to following.
1. You mention that you used 4 strains of Gymnodinium catenatum. You got these strains of CODIMAR. Why not providing a microscopy good magnification and resolution photo of each strain? This kind of studies are read by students and researchers of other fields not accustomed necessarily with these organisms. So an array of 4 photos in a plate will much add in value of this manuscript.
2. You cultured them. The culture in itself is of paramount importance for your study. It is the core of it. But you offer few things in your presentation. What was the duration of the cultures? How did you calculate SGR? Which days you used for calculation of SGR? How can we figure out the initial phase, log phase, stationary phase, without a graph? You must provide a graph with the growth curves of each culture for the duration of the cultures.
3. For the determination of cell density as cells/mL you used a Sedwick-Rafter chamber. So, you did not use any haematocytometer. Is that possible? What is the size of Gymnodinium? That's why you should provide information (along with the photos) of this dinoflagellate. From my personal experience even with large protists of ~ 20-30 μm the most accurate and reliable method is the haematocytometer method. I really do not feel comfortable evaluating your method.
4. Table 2. What fmol/cell means? Not everyone is accustomed with fmol. In the caption of Table 2 you mention growth rate, max. cell density and toxicity. Where are these in the content of the table?
5. Table 1. may be referred to the content of Table 2. Its a mess. Fix it.
6. Line 169 correct the value of 0.820.7 div day-1.
7. In Line 98 you start to mention the term "toxicity". Toxins, I can understand, but toxicity on what? Did you perform any toxicity test on some kind of organisms?
8. In Figures 1, 2, 3 and 5 what the error bars stand for? Are they SD or SE? Define. The letters over the bars indicate stat. differences as you write. It means that all "a" are equal, all "b" equal etc. right? If it is e.g. "a,b" it means that a and b are statistically equal, right? Looking the lettering at all figures it looks that by no means all the statistics you depict are correct. Something wrong is in your figures. Amend them thoroughly.
9. In Lines 34, 36, 49, 51, 56 and elsewhere (maybe) the arrangement of the numbers in the brackets are not ordered. Amend it.
10. In conclusion section you should define clearly the take-away message for the readers. In that sense which is the nitrogen concentration that generates HAB? Also I do not understand how the N:P ratios do not affect growth and toxins while N-concentration generate HAB? Again what toxicity refers to? My general impression is that your conclusion left me perplexed. You should rewrite the conclusions.
Reviewer 2 Report
In general, the manuscript is well written and organized. However, the data in the figures does not match the data detailed in the text in several instances. This lack of scientific rigor is the major drawback of this scientific report.
Specific comments:
Line 30 – the common abbreviation for these toxins is PSTs (paralytic shellfish poisoning toxins), not PTs. There are diverse aquatic neurotoxins, so it’s best to keep this widely used abbreviation.
Line 34- the list of references was not ordered in ascending order. Same comment for lines 36 and 41.
Line 82-«BAPAZ-5 maintained an average cell density of 498 ± 275 and 2814 ± 212 cells mL -1 with N:P ratios of 8:1 and 4:1» This is not correct. In Fig. 1b, both presented cell densities around 2500, and a non-significant difference (‘a’).
Line 92 – strain BAMAX2 only presented significantly different average cell abundance between N:P ratios of 64:1, not 32:1, according to Fig. 1b.
Line 94- there was no statistical letter assigned to BAPAZ5 in Fig. 1b.
Fig 2. – the graphs present different scales: it is difficult to compare the same strain between different treatments. I suggest presenting Fig. 2 like Fig. 1.
Line 115- «maximum toxicity of BAMAZ-2 (28.65 ± 4.36 pg cell-1 ) was registered with the N:P ratio of 4:1» There is a serious lack of scientific rigor here: Figs. 2d and 2e show toxicity of BAMAZ-2 up to 60 pg!
Line 122-«BAPAZ-7 showed higher toxicity with N:P ratios ranging from 16:1 (25.48 ± 5.98) to 64:1 (38 ± 7.34 pg cell-1 ; Figures 3 D, E)» Another serious lack of scientific rigor here: Fig. 2e show toxicity of BAMAZ-2 up to 150 pg, the graph even presents a broken line to fit this scale!
Fig 3. – the graphs present different scales: it is difficult to compare the same strain between different treatments. I suggest presenting Fig.3 like Fig. 1, maybe separated by toxin group in a,b,c graphs.
Table 3- for strain ‘GCMQ-4’ no info was provided on ‘date of isolation’ nor ‘water temperature’, also, the footnote ‘nd’ was not included in the case of ‘missing data’.
In the abstract and the text, it was not clearly explained that the different N:P ratios were obtained with a fixed P concentration, varying only the nitrogen.
Reviewer 3 Report
This interesting study is devoted to the monitoring for growth, toxin profiles, and toxin content of four Gymnodinium catenatum strains isolated from Bahía de La Paz (BAPAZ) and Bahía de Mazatlán (BAMAZ) depending on different N:P ratios.
The authors sought to improve their understanding of the ecophysiology of G. catenatum by determining the effect of different ratios of nitrogen: phosphorus concentrations on cell density, growth rate, toxicity and toxin profile of the tested strains.
There are several comments aimed at improving the clarity of the text.
From this work, we learn that the BAPAZ and BAMAZ strains of G. catenatum did not differ significantly in growth rate, toxin production and toxin profile. However, the growth rates and the number of cells were mainly influenced by the specific concentrations of the nitrogen source (NaNO3).
The reader will ask the questions: "How are the growth rate and the number of cells related? Why do different ratios of nitrogen and phosphorus affect these parameters so much? Why such ratios of nitrogen and phosphorus were chosen for the study?"
A general summary Table could help authors and readers analyze the results obtained.
The authors should discuss their results, explaining why such effects are observed. What is probably known about the behavior of other strains under similar growth conditions described in the literature? Why is nitrogen so important for the tested parameters? How can this be related to the N:P ratios? What cellular processes can be influenced by different ratios of nitrogen and phosphorus in the growth medium?
It is necessary to make a clearer and more specific formulation of the conclusions.
Interesting data were obtained in the work. However, this manuscript needs revision. The text lacks clarity, good discussion, and clear concrete conclusions.
Round 2
Reviewer 1 Report
In my first review I quoted:
In Figures 1, 2, 3 and 5 what the error bars stand for? Are they SD or SE? Define. The letters over the bars indicate stat. differences as you write. It means that all "a" are equal, all "b" equal etc. right? If it is e.g. "a,b" it means that a and b are statistically equal, right? Looking the lettering at all figures it looks that by no means all the statistics you depict are correct. Something wrong is in your figures. Amend them thoroughly.
You replied:
The error bars indicate SD, this was clarified in each figure. We used a one-way ANOVA to compare each strain within each N:P ratio (eg 4:1), for each N:P ratio between strains, not between N:P ratios. Columns with the same letter indicate they do not differ significantly.
You clarified that error bars stand for SD. O,K. Lets proceed to indicators of statistical difference at the top of the bars. You wrote that they represent comparisons between strains within each N:P ratio. O.K. So you have 5 N:P ratios for a set of 4 strains. Lets take the case of the 8:1 ratio in the lower graph of abundance in Figure 1. You put the superscripts of a, bc, a and c above each bar of each strain. Lets interpret it. You mean that a and a are statistically equal? You mean bc and c are equal between b and c? and that b is allocated to BAMAZ-2? How is it possible GCMQ-4 (a) and BAMAZ-5 (a) be statistically equal? Even a simple glance at the figure "cries" to you that they are different (are you sure that Tukey's result after you performed one-way ANOVA indicates equality?}. I think that all the confusion (in all figures) stems from the fact of the poor method of depiction of the letters over the bars. I suggest the following safe, clear and concise method of lettering. In each bar representing each strain you put a different letter, a, b, c, d. Then for statistically equal bars you add the relevant letter next to its unique one. For example after you assigned a, b, c, d to the bars, if (e.g.) a, b and c are equal then over each bar should be: (a), (bac), (cab), (d). Only by this way a very informative graph can be robust. And please perform your statistics again just to be sure. Another tip is to use SE bars instead of SD bars. They are more informative. But this is not compulsory, do it if you accept mu advice.
All other amendments are acceptable even if I strongly disagree with your statement that it is difficult to count by haematocytometer large protists of 30-40 μm. Using 100X magnification it is very easy and accurate.
Reviewer 2 Report
R0: Fig. 1b - strain BAMAX2 only presented significantly different average cell abundance between N:P ratios of 64:1, not 32:1, according to Fig. 1b. // R1: Despite the answer, all the ratios present letter ‘b’, only in ratio 64:1 there is letter ‘a’. So the new caption does not solve the difficulty in interpretation.
Section 2.3. - Regarding toxin content x toxicity, it can be confusing sometimes. Now the authors have improved this deference.
